# Oscillatory integration windows in neurons

Nitin Gupta[1,2], Swikriti Saran Singh[2] & Mark Stopfer[1]

Oscillatory synchrony among neurons occurs in many species and brain areas, and has been proposed to help neural circuits process information. One hypothesis states that oscillatory input creates cyclic integration windows: specific times in each oscillatory cycle when postsynaptic neurons become especially responsive to inputs. With paired local field potential (LFP) and intracellular recordings and controlled stimulus manipulations we directly test this idea in the locust olfactory system. We find that inputs arriving in Kenyon cells (KCs) sum most effectively in a preferred window of the oscillation cycle. With a computational model, we show that the non-uniform structure of noise in the membrane potential helps mediate this process. Further experiments performed *in vivo* demonstrate that integration windows can form in the absence of inhibition and at a broad range of oscillation frequencies. Our results reveal how a fundamental coincidence-detection mechanism in a neural circuit functions to decode temporally organized spiking.

[1] National Institute of Child Health and Human Development, National Institutes of Health, Bethesda, Maryland 20892, USA. [2] Department of Biological Sciences and Bioengineering, Indian Institute of Technology Kanpur, Kanpur 208016, India. Correspondence and requests for materials should be addressed to N.G. (email: guptan@iitk.ac.in) or to M.S. (email: stopferm@mail.nih.gov).

Oscillatory synchronization of neurons occurs in many brain regions[1], including the olfactory systems of vertebrates[2,3] and invertebrates[4–7], and is indispensable for precise olfactory coding[5,8]. One mechanism by which oscillations have been proposed to influence coding is through the creation of cyclic integration windows—specific times within the oscillation cycle when synaptic input is most efficiently integrated by a postsynaptic neuron[9]. Cyclic integration windows could allow a neuron to respond preferentially to spikes arriving from multiple presynaptic neurons coincidentally in a specific part of the cycle[10]. Thus, coincidence detection mediated by integration windows could help read precise temporal codes for odours[10,11]. Phase-specific effects of synaptic inputs have been described both in brain slices[12,13] and in simulations[14,15]. However, the existence of cyclic integration windows has not been demonstrated, and their functional requirements are unknown.

We examined cyclic integration windows in the locust olfactory system. Here, information about odours is carried from the antenna to ∼800 projection neurons (PNs) in the antennal lobe. Individual PNs respond to multiple odours with dense, time-varying patterns of spikes[9]. Excitatory and inhibitory interactions between PNs and local neurons in the antennal lobe tend to synchronize subsets of PNs so their spikes arise in waves of ∼20 Hz oscillations[9]. PNs carry the synchronized spikes to the mushroom body, where each of the 50,000 Kenyon cells (KCs) receives input from a subset of PNs (ref. 16). KCs have been described as coincidence detectors[10], responding selectively to odours, with very few spikes. In each oscillatory cycle, KCs receive

excitatory input from PNs followed by inhibitory input from GABAergic neurons[10,17]. Perez-Orive et al.[10] suggested this inhibition could periodically antagonize the excitation, leaving a brief time window in every cycle, between the arrivals of excitatory and inhibitory inputs, for KCs to efficiently integrate the input. Thus, not only would KCs receive more presynaptic inputs within the window, but each input occurring within the window would contribute more to the firing of the KC than an equally strong input occurring outside the window[10]. Cyclic integration windows therefore provide a possible mechanism for amplifying and privileging synchronized inputs from PNs. Spikes generated by KCs strongly phase-lock with the oscillations[6,10], but whether this occurs because KCs integrate input more effectively at some phases determined by integration windows, or simply because KCs receive more input at those phases, is not known.

We sought to determine whether and how the cyclic input arriving in KCs establishes integration windows in these neurons. In awake animals we elicited oscillatory drive to KCs by delivering odours to the antenna, and then manipulated excitatory inputs directly to KCs by injecting pairs of current pulses. With simultaneous intracellular and field potential recordings we found that summation of inputs was more efficient in a specific part of the oscillatory cycle, demonstrating effective integration windows in KCs. Next, with a computational model we revealed the substantial inhibition-independent contribution of noisy membrane potential fluctuations to this process. To test further predictions of the model we then directly injected purely excitatory oscillatory input to KCs, establishing that integration

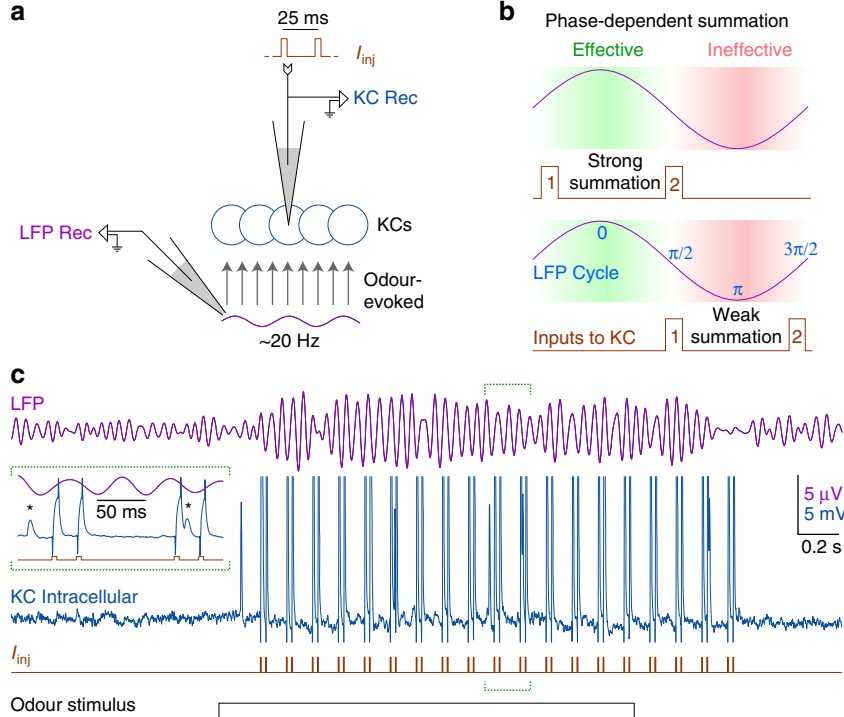

**Figure 1 | Experiment design. (a)** Recording configuration: In locusts, oscillating waves of odour-evoked spikes are carried by projection neurons from the antennal lobe to Kenyon cells (KCs). A blunt electrode in the mushroom body monitored the odour-evoked oscillatory local field potential (LFP Rec) while an intracellular electrode in a KC (KC Rec) recorded membrane potential and injected pulses of current ($I_{inj}$). **(b)** Experiment strategy: By injecting pairs of current pulses into KCs (upper and lower panels show two such pairs) we tested the effect of phase position on input summation. **(c)** Example experiment from $n = 16$ KCs: LFP oscillations (purple, top) were induced by a 2-s odour presentation (black, bottom). The second trace (blue) shows a simultaneous intracellular recording from a KC. In each trial, the KC received 19 pairs of intracellular current pulses ($I_{inj}$, red) with the following parameters: pulse-width = 5 ms; inter-pulse interval = 25 ms (onset-to-onset); inter-pair interval = 125 ms (onset-to-onset). Each neuron received at least 20 trials. Inset: A 210-ms period spanning the 10th and the 11th pulse pairs. Although current injections caused large artefacts in the recording, spikes (marked with asterisks) could be identified between or superimposed with the stereotyped artefacts.

windows can be formed without inhibition, and showing that integration windows arise despite variations in oscillation frequency and interpulse interval. Our results establish that integration windows exist and provide a robust, simple and flexible mechanism for coincidence detection using oscillations.

## Results

**Summation in KCs is regulated by cyclic integration windows.** In intact and awake locusts we made intracellular recordings from a KC ($n = 16$) while simultaneously recording odour-evoked LFP oscillations ($\sim 20$ Hz) with an extracellular electrode in the mushroom body (Fig. 1a). We injected pairs of identical intracellular current pulses into the KC and tested whether the summation of the two pulses depended on the phase of the oscillations (Fig. 1b). Intracellular current injections allowed us to control, with millisecond precision, the time when the input reached the KC, while minimizing indirect inputs. Current pulses arrived at different phases owing to variations in the onset and duration of the odour-elicited oscillatory cycles. Pulses in each pair were separated by 25 ms ($\sim 1/2$ cycle) so the two pulses would be equally likely to fall within or between cycles. The interval between pulse pairs was 125 ms, sufficient for the membrane potential to return to baseline before the next pulse pair arrived (Fig. 1c).

Consistent with expectations and earlier results[4], we found that odours evoked oscillations in the subthreshold membrane potential of KCs (Fig. 2a), with depolarizing (upward) deflections in the first half of the cycle. The average peak-to-peak amplitude of these oscillations was $1.1 \pm 0.7$ mV ($n = 16$ cells). We characterized the membrane properties of the recorded KCs by their responses to current injections: the input resistance in the recorded KCs ($n = 16$) varied between 151 and 1,082 MΩ; net capacitance varied between 5.8 and 40.1 pF; and membrane time constant varied between 2.3 and 15.2 ms. Current pulses induced depolarizations that decayed over a few tens of milliseconds (Fig. 2b). Twenty to 25 ms after a pulse (just before the beginning of the second pulse), we observed $1.21 \pm 0.44$ mV of residual depolarization. The magnitude of the residual depolarization was expected to be independent of the phases at which the pulses arrived, which we confirmed (Pearson correlation between depolarization and membrane potential oscillations as functions of phase, $r = 0.32$, $P = 0.31$; Fig. 2c).

To investigate the integrative properties of the KCs we measured the probabilities of spiking in response to the first pulse, $R_1(\phi)$, and to the second pulse, $R_2(\phi)$, as functions of their phase, $\phi$, with respect to the LFP oscillation cycle. In the presence of odour, the membrane potential and the spikes of KCs were expected to show oscillations that phase-locked with the LFP (refs 4,10). Also, because an input at a depolarized phase was expected to elicit more spiking than the same input at a hyperpolarized phase, $R_1(\phi)$ and $R_2(\phi)$ were also expected to show similar phase-locking. Indeed, both $R_1(\phi)$ and $R_2(\phi)$ were phase-locked to the LFP (Fig. 2d) and were highly correlated with the membrane potential oscillations (Pearson correlation, $r = 0.74$, $P = 0.006$ for $R_1(\phi)$; $r = 0.79$, $P = 0.002$ for $R_2(\phi)$; $n = 12$ phase-bins per correlation).

As the first and second pulses were identical in magnitude, spiking generated by both should be equally dependent on the membrane potential oscillations. Therefore, subtracting $R_1(\phi)$ from $R_2(\phi)$ for any given $\phi$ controls for the direct contributions of oscillations on the spiking generated by an input. Thus, $R_2(\phi) - R_1(\phi)$ quantifies the phase dependency of any excess spiking caused by the summation of the two pulses. In the absence of any summation between the first and the second pulses, $R_2(\phi) - R_1(\phi)$ will be 0. Note that the two pulses in a pair arrive at different phases and so cannot be directly compared (see Fig. 1b); thus, for

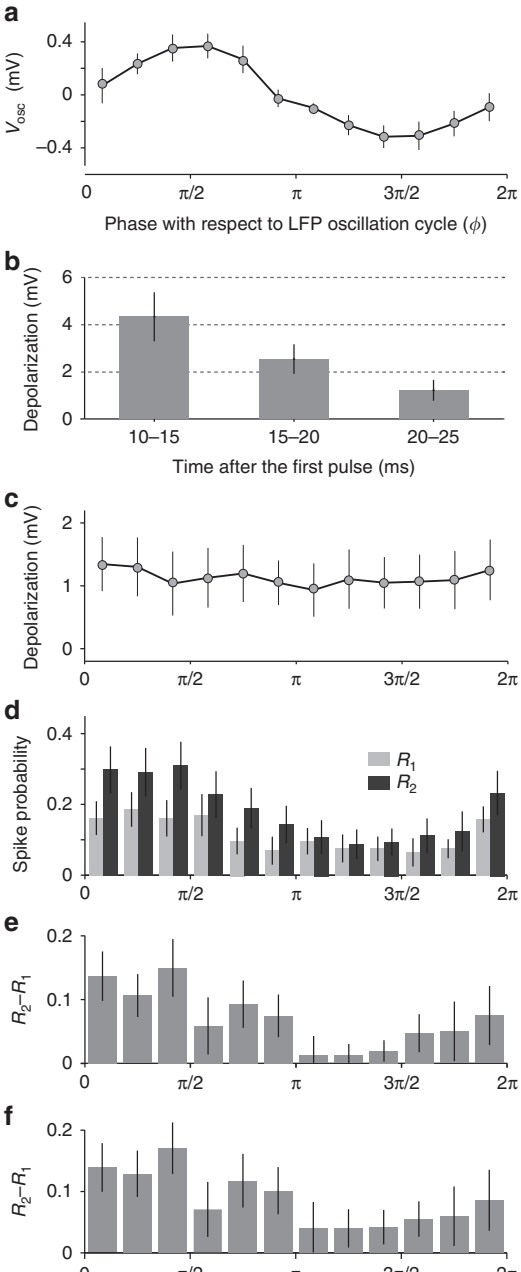

**Figure 2 | KCs have cyclic integration windows.** (**a**) Average membrane potential plotted as a function of the LFP phase ($\phi$) shows robust subthreshold oscillations ($V_{osc}$) with upward deflections in the first half of the cycle; $n = 16$ KCs, error bars: s.e.m. (**b**) Residual depolarization in different time intervals after the onset of the first pulse (the pulse ends at 5 ms). (**c**) Residual depolarization in the 20–25 ms interval after the first pulse (that is, just before the onset of the second pulse) did not depend on the phase of the oscillation cycle. (**d**) Spiking responses (mean ± s.e.m.) evoked by the first and the second current pulses, $R_1(\phi)$ and $R_2(\phi)$, respectively, are most likely to occur in the first half of the cycle. (**e**) Phase-dependent summation is measured by comparing responses to the first pulse ($R_1(\phi)$) and the second pulse ($R_2(\phi)$) occurring at the same phase (not to the two pulses in the same pair). The effect of summation, $R_2(\phi) - R_1(\phi)$, is most effective in the first half of the cycle. (**f**) To test possible contributions of after-hyperpolarizations to phase-dependent summation, $R_2(\phi) - R_1(\phi)$ was computed after excluding responses to pairs of pulses in which the first pulse evoked a spike. This data set lacking after-hyperpolarizations also showed robust phase-dependent summation. All error bars represent s.e.m.

each phase position $\phi$, we measured the effect of summation as the difference in average responses to all first pulses at $\phi$ [$R_1(\phi)$] and all second pulses at $\phi$ [$R_2(\phi)$] after collecting responses from all pairs.

Unexpectedly, $R_2(\phi) - R_1(\phi)$ showed a strong phase preference: summation was significantly more effective when the second pulse occurred in the first (rising) half of the oscillation cycle (t(10) = 3.78, $P = 0.004$, two-sample $t$ test, $n = 6$ phase-bins per sample), and the amount of summation was correlated with the membrane potential oscillations ($r = 0.70$, $P = 0.01$; Fig. 2e). This was unexpected for two reasons; first, although $R_1(\phi)$ and $R_2(\phi)$ were both greater in the first half of the cycle (when membrane potential is depolarized), this 'direct' effect of depolarization had been removed by subtracting $R_1(\phi)$ from $R_2(\phi)$; second, $R_2(\phi) - R_1(\phi)$ was phase-dependent even though residual depolarization from the first pulse was demonstrably not phase-dependent (Fig. 2c). Notably, the unexpected effect of phase-dependent summation was just as strong as the 'direct' effect of phasic depolarization alone (maximum difference of 0.13 in $R_2 - R_1$ values across phases, Fig. 2e, compared to 0.12 for $R_1$ across phases, Fig. 2d). These results provide the first evidence for phase-dependent summation in neurons. The positive correlation between membrane potential oscillations and summation efficiency establishes not only that KCs receive more inputs (from PNs) in a particular window of the oscillation cycle but also that each input received in that window is more effective. These results show that integration of inputs is more effective in a specific window of the oscillation cycle, thus demonstrating effective cyclic integration windows in KCs.

**Membrane potential noise contributes to integration windows.** It was not immediately clear why summation should be phase-dependent. Notably, it was not mediated simply by the presence of oscillatory depolarization. Then, what mechanisms underlie phase-dependent summation? We first considered the possibility that the after-hyperpolarization following a spike elicited by the first injected current pulse suppresses the response to the second pulse. Our reasoning was this: if the first pulse arrives near the peak of the membrane potential oscillation, it would have a high probability of eliciting a spike. Then, this spike's after-hyperpolarization could interfere with the response to the second pulse, which arrives 25 ms later. Thus, a second pulse that arrives nearer the trough (in case of 50-ms oscillation cycle) would be less likely to elicit a spike than a first pulse at the same phase. This timing relationship could potentially contribute to the correlation we observed between $R_2 - R_1$ and membrane potential oscillation. To test whether this relationship is responsible for the phase-dependent summation we reanalysed our results, now including only those pulse pairs in which the first pulse did not elicit a spike; thus, in this analysis, no after-hyperpolarizations followed the first pulses. However, in these cases we still observed phase-dependent summation (Fig. 2f): $R_2(\phi) - R_1(\phi)$ was greater in the first half of the cycle (t(10) = 4.32, $P = 0.002$, $t$ test), and correlated significantly with the membrane potential oscillations ($r = 0.74$, $P = 0.006$). These results rule out an essential contribution of after-hyperpolarizations to phase-dependent summation and suggest a different mechanism is responsible.

Even in the absence of odour or current inputs, intracellular recordings showed noisy membrane potential fluctuations reflecting, in part, subthreshold synaptic activity (see, for example, in Fig. 1c, the first two seconds of each trial)[18]. The average amplitude of this noise was 0.82 mV (standard deviation of the membrane potential), 4.78 mV (peak-to-peak amplitude). This input could affect the probability of the KC reaching

spiking threshold. To test this and to investigate the minimal requirements for phase-dependent summation, we constructed a simple computational model of a KC (see Methods). The membrane potential ($V$) was determined by three independent factors based on measurements we had made *in vivo*: cyclic excitatory input from PNs, which generated subthreshold oscillations ($V_{osc}$) of amplitude $\sim 1.5$ mV; a noise term reflecting fluctuations in $V$; and injected excitatory current pulses ($I_{inj}$) (Fig. 3a). We found that, consistent with experimental results, responses of the model neuron to single pulses, $R_1(\phi)$, depended strongly on phase, and correlated with subthreshold oscillations in the membrane potential ($r = 0.94$, $P = 4.0 \times 10^{-6}$; Fig. 3b,c). Notably, even this simple model showed phase-dependent summation of paired current pulses: $R_2(\phi) - R_1(\phi)$ correlated strongly with subthreshold oscillations ($r = 0.92$, $P = 2.5 \times 10^{-5}$). The results were robust to variations in key model parameters, such as the amplitude of the injected current pulses, the amplitude of the subthreshold oscillations and the amplitude of the noise term in membrane potential (Fig. 3d).

A comprehensive analysis of synaptic noise required considering the shape of the noise distribution. Figure 3e illustrates the relationships among the membrane potential, the distribution of noise and the probability of spiking. In the absence of noise and current pulses, $V(\phi) = V_{osc}(\phi)$. When noise is added, $V(\phi)$ follows a Gaussian distribution with mean $V_{osc}(\phi)$. Even if the source of the noise is not Gaussian, the cumulative effect of noise on the membrane potential approaches a Gaussian distribution (central limit theorem). The area of this distribution above the spiking threshold equals the probability of spiking at phase $\phi$. Injecting a positive current pulse shifts this distribution upward by raising the mean, increasing the probability of spiking. If a pair of current pulses is injected, the second pulse shifts the distribution of $V$ further upward by residual depolarization $\Delta V$ (the depolarization elicited by the first pulse remaining when the second pulse arrives). This upward shift also increases the area of the distribution above the threshold, from $R_1(\phi)$ to $R_2(\phi)$. Because the constitutive membrane properties of the cell and the delay between the two pulses are constant, $\Delta V$ is independent of $\phi$ (Fig. 2c shows this is true *in vivo*). However, notably, the non-uniform, bell-like shape of the distribution causes $R_2(\phi) - R_1(\phi)$ to depend on $\phi$; thus, $R_2 - R_1$ is greatest during the phase with the most spiking (Fig. 3e, top).

The analysis shown in Fig. 3e suggests that summation will be independent of phase if the area under the curve between two thresholds separated by $\Delta V$ were independent of $V$, a condition that is guaranteed only for a uniform distribution. To test this prediction, we artificially set the noise distribution in the model to be uniform (see Methods) instead of Gaussian. As predicted, summation in the presence of a uniform distribution became independent of phase ($r = 0.13$, $P = 0.68$; Fig. 3f). Thus, our model revealed that the inevitable Gaussian, nonlinear shape of the noise distribution underlies phase-dependent summation.

The computational analysis showed that, in principle, phase-dependent summation could occur even when noise amplitude was low (Fig. 3d right). However, further analysis of the model (Supplementary Fig. 1) revealed that this occurs when $R_1(\phi) = 0$ but $R_2(\phi) > 0$ for some $\phi$ (that is, when a single input never elicits spiking at phase $\phi$ but paired inputs do). Our recordings made *in vivo* (Fig. 2) showed that $R_1(\phi)$ was always $> 0$ for all phases. Thus, phase-dependent summation observed in KCs could not be explained by this low noise effect.

The computational model gave rise to three additional predictions that we describe in the following sections: integration windows should not require inhibitory input, should be robust to variations in oscillation frequency and should be robust to

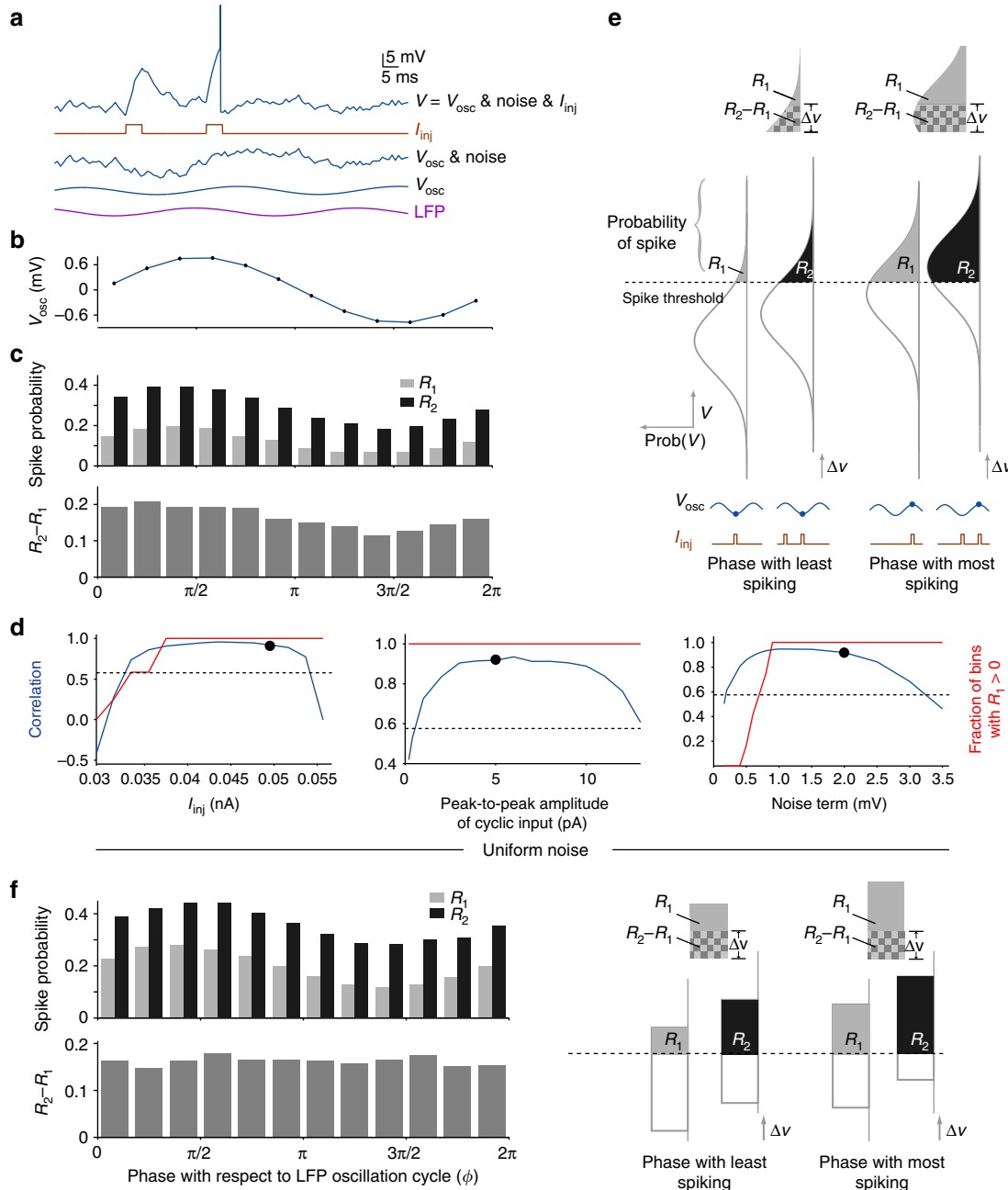

**Figure 3 | Computational model shows noise contributes to phase-dependent summation.** (**a**) Top: representative membrane potential trace from a model KC. The membrane potential ($V$) has three independent components: subthreshold oscillations ($V_{osc}$) generated by the integration of oscillatory input from PNs; noise and a pair of current pulses ($I_{inj}$) injected into the cell. In this example, the second current pulse elicited a spike. Oscillations in both LFP and $V_{osc}$ are caused by the same oscillatory input from PNs, but are delayed in $V_{osc}$ (see Methods). (**b**) The model's average membrane potential as a function of the LFP phase ($\phi$). (**c**) $R_1(\phi)$ and $R_2(\phi)$ show a phase-dependency similar to $V_{osc}$. The effect of summation, $R_2(\phi) - R_1(\phi)$, also shows a similar phase-dependency. (**d**) Blue lines: correlation between $R_2$–$R_1$ and subthreshold oscillations was robust to variations in model parameters, such as the amplitude of the injected current pulses ($I_{inj}$), the amplitude of the subthreshold oscillations, and the amplitude of noise. Dots: parameter values used in simulations shown in **a**-**c**. Dashed line: correlations larger than this value (0.576) are statistically significant ($P < 0.05$). Red line: fraction of phase bins (of 12) in which $R_1(\phi) > 0$. (**e**) A graphical description of phase-dependent summation in the presence of noise. The membrane potential oscillates ($V_{osc}$) when given cyclic input. Noise causes the actual voltage $V$ at a given phase to vary from the mean. The probability distribution of $V$ at any phase must be approximately Gaussian because of the accumulation of noise. The area of this distribution above the spiking threshold equals the probability of spiking at that phase. If the cell receives two current pulses, the distribution of $V$ during the second pulse shifts upward by $\Delta V$ relative to the distribution of $V$ during the first pulse. At the phase where $V_{osc}$ is lowest (left), shifting the distribution by $\Delta V$ produces a small change in area above the curve ($R_2 - R_1$, shown in checkerboard pattern). At the phase where $V_{osc}$ is highest (right), the same shift $\Delta V$ produces a larger change in the area (compare the two checkerboard areas). Thus, owing to the nonlinear shape of the noise distribution, the first pulse makes a larger contribution to the second pulse when the second pulse occurs at phases close to the peak of $V_{osc}$. (**f**) If the noise distribution is made uniform (see Methods), $R_2 - R_1$ loses its phase-dependence and correlation with subthreshold oscillations.

variations in interstimulus interval. We tested these predictions *in vivo*.

**Integration windows can form in the absence of inhibition**. The odour-evoked oscillatory excitation that KCs receive from PNs is accompanied by feedback inhibition from another network element, a giant GABAergic neuron[19–21]. This inhibition is also phase-locked to the oscillations[10,17]. Indeed, integration windows in KCs were originally proposed to be sculpted by the phase-locked inhibition[10]. However, results from our simple computational model, which lacked inhibitory input, suggest that purely excitatory oscillatory input suffices to generate effective integration windows. These results lead to the prediction that KCs will exhibit integration windows when provided with oscillatory input even in the absence of inhibition.

To test this prediction, we used direct intracellular current injections (see Methods) to individual KCs to provide purely depolarizing oscillatory input, along with the usual pairs of excitatory current pulses. Thus, the input included no inhibition (Fig. 4a,b). In each experiment ($n = 10$) current was injected into a single KC, one of nearly 50,000 in each hemisphere[4]. Because feedback inhibition to KCs is proportional to the net activity of the whole KC population[19], our stimulation of a single KC should generate negligible inhibitory feedback.

Current injections successfully generated 20-Hz oscillations in the membrane potential (Fig. 4a,c) with an average peak-to-peak amplitude of $1.24 \pm 0.58$ mV. These oscillations affected both

$R_1(\phi)$ and $R_2(\phi)$ reliably, and resulted in phase-dependent summation similar to that evoked by odour delivery: $R_2(\phi) - R_1(\phi)$ was stronger in the first half of the cycle ($t(10) = 3.69$, $P = 0.004$, $t$ test), and correlated with the membrane potential oscillations ($r = 0.82$, $P = 0.001$; Fig. 4c). These results show that effective integration windows can be formed without inhibition.

**Integration windows are robust to oscillation frequency**. Are mushroom bodies hard-wired to resonate at $\sim 20$ Hz (ref. 22), or, as our computational model predicted (Fig. 5a), is the integration mechanism sufficiently flexible to operate at other frequencies? If integration windows are determined solely by oscillatory input, it should be possible to form integration windows given inputs at another frequency. Using direct current injections (Fig. 5b), we provided oscillatory depolarizing input to KCs with an arbitrary frequency of $\sim 12$ Hz (84-ms period; see Methods), and tested KCs for the emergence of integration windows.

KCs exhibited robust subthreshold oscillations at $\sim 12$ Hz, with an amplitude of $1.29 \pm 0.49$ mV ($n = 10$ cells). Thus, KCs were able to follow oscillatory input at an arbitrary frequency as strongly as they follow the typical odour-elicited frequency of 20 Hz. Further, KCs showed robust integration windows at the tested frequency: $R_2(\phi) - R_1(\phi)$ was stronger in the first half of the cycle ($t(10) = 3.56$, $P = 0.005$, $t$ test), and correlated with the membrane potential oscillations ($r = 0.64$, $P = 0.02$; Fig. 5c). Therefore, effective integration windows can be formed by oscillatory input even at frequencies not generated by olfactory

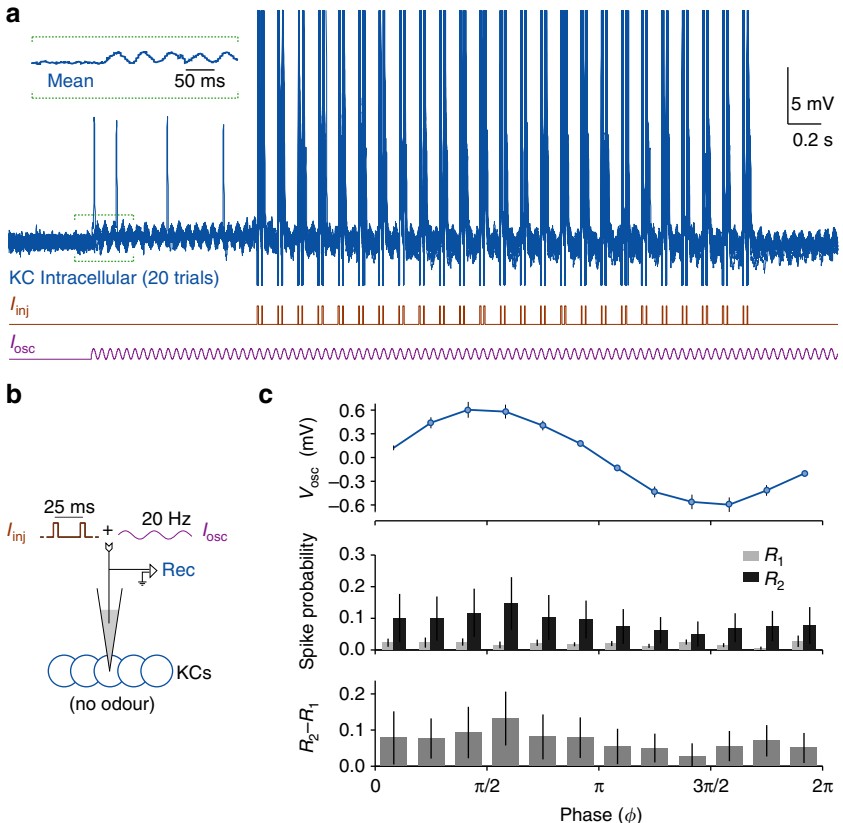

**Figure 4 | Integration windows can form in the absence of inhibition.** (**a**) A representative experiment from $n = 10$ KCs, and (**b**) the experiment design are shown. Purely excitatory oscillatory waveforms ($I_{osc}$), and 25 pairs of 5-ms pulses ($I_{inj}$), were injected directly into the KC through the intracellular electrode, without eliciting network inhibition. Membrane potential oscillations were induced by $I_{osc}$. Inset: the mean membrane potential (20 trials) of the indicated segment. (**c**) Current injections evoked robust oscillations in the subthreshold membrane potential of the KC. Lacking inhibition, KCs still formed integration windows: $R_2 - R_1$ showed the same phase-dependence as seen previously with odour stimulation (see Fig. 2). Error bars represent s.e.m.

circuitry or typically encountered during odour presentations. Further, the observation that KCs can form integration windows at an arbitrary frequency confirms that cyclic integration windows can be driven by oscillatory input alone, and do not necessarily require frequency-tuned intrinsic properties like resonance, or inhibition.

**Integration windows are robust to interstimulus interval**. The computational model suggested that integration windows should be effective over a range of interstimulus intervals (Fig. 6a). To test this *in vivo* we modified the experiment illustrated in Fig. 2, and, using odour puffs to elicit ∼20 Hz oscillations, we delivered pulse pairs with short (15 ms) or long (35 ms) separations (Fig. 6b). With 15-ms separation ($n = 18$ KCs), $R_2(\phi) - R_1(\phi)$ was again greater in the first half of the cycle (t(10) = 2.50, $P = 0.03$, $t$ test), and significantly correlated with the membrane potential oscillations ($r = 0.82$, $P = 0.001$; Fig. 6c). Similarly, with 35-ms separation ($n = 13$ KCs), $R_2(\phi) - R_1(\phi)$ was stronger in the first half of the cycle (t(10) = 6.39, $P = 0.00008$, $t$ test), and significantly correlated with the membrane potential oscillations ($r = 0.70$, $P = 0.01$; Fig. 6d). These results confirm the phase-dependent summation in KCs shown in Fig. 2 and further show that this effect is not limited to inputs separated by a specific interval.

## Discussion

Our results provide the first demonstration of cyclic integration windows in neurons: we found that the summation of two nearly coincident current injections in KCs is more effective during a specific, narrow portion of the oscillation cycle (Fig. 2). We also

studied the mechanisms underlying the origin of integration windows. Our computational model suggested this feature is elicited, in large part, by oscillatory input and the inevitable, non-uniform shape of the synaptic noise distribution of the membrane potential (Fig. 3). Using direct injections of oscillatory depolarizing current into KCs, we verified that integration windows can arise in the absence of inhibition (Fig. 4). Further confirming that integration windows are driven by the oscillatory input rather than, for example, resonance properties, we showed that KCs can form integration windows from input at an arbitrary frequency (Fig. 5). And consistent with a prediction of our computational model, phase-dependent summation was robust to broad variations in interstimulus interval (Fig. 6).

These results are important for understanding how temporal codes for odours[23–25] may be decoded. Phase-dependent summation allows individual spikes of PNs to have variable importance to their downstream followers, the KCs (ref. 9); PN spikes that arrive at KCs within a preferred window of the oscillation cycle can be integrated with other near-coincident spikes more effectively than spikes that arrive outside the window. Thus, integration windows provide a mechanism for reading out information contained in the precise timing of spikes[26–28]. Integration windows also provide a mechanism for propagating synchrony across layers of neurons: integration windows reinforce phase-locked spiking in neurons by increasing their sensitivity to inputs at favoured phases; phase-locked spiking, in turn, creates integration windows in the next layer of neurons.

Our results establish that cyclic integration windows can be formed from very few ingredients: oscillatory input and noisy fluctuations in the membrane potential. Given the ubiquity of

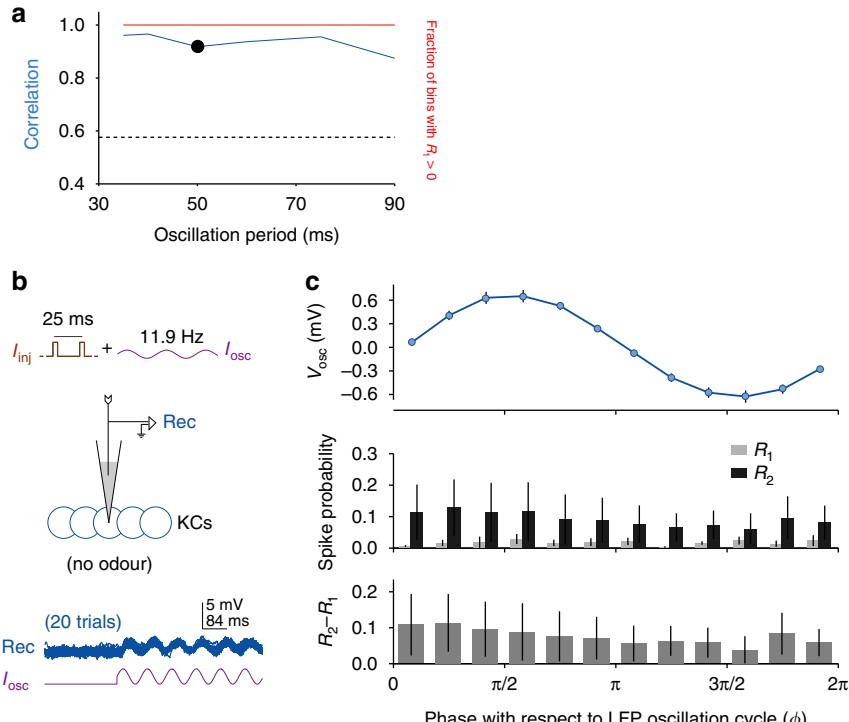

**Figure 5 | KCs form integration windows robustly even at frequencies other than 20 Hz.** (**a**) The computational model predicted phase-dependent summation for a wide range of oscillation cycle periods (this panel uses the same format as Fig. 3d). (**b**) *In vivo* test: experiment design is similar to the one shown in Fig. 4: Oscillatory input with an arbitrary frequency of 11.9 Hz (84-ms period) was injected into the KC along with pairs of current pulses. Bottom: membrane potential oscillations generated by the current in a representative neuron (first 500 ms of current injection is shown), from $n = 10$ KCs. (**c**) Current injections at 11.9 Hz generated robust oscillations in the subthreshold membrane potential of the KC. KCs exhibited integration windows: $R_2 - R_1$ showed the same phase-dependence as seen previously with 20-Hz oscillations (Fig. 2). Error bars represent s.e.m.

membrane noise, the mechanisms we describe likely apply to a wide variety of neurons that receive oscillatory inputs[1], with or without inhibition and across a range of frequencies (Figs 4 and 5). Further, the ability of a given type of cell to form integration windows at multiple frequencies could allow robust oscillation-based coding despite frequency changes caused by stimulus variations[29], stimulus duration[6,30], or varying internal states of the animal (reviewed in ref. 31).

Our results show that inhibition and active conductances are not needed to form integration windows. However, coincidence detection by KCs may be enhanced by their active conductances and phase-locked inhibitory input from other neurons[32,33]. KCs were originally proposed to receive feedforward inhibition through the lateral horn[10], but recent work has shown that the inhibition originates mostly from the unique feedback neuron giant GABAergic neuron (refs 17,19,20). Because this inhibitory feedback is also phase-locked to the LFP (ref. 17), it may contribute to integration windows in KCs.

The sparse responses of KCs may facilitate association of odours with reward[9,34], consistent with a role for the mushroom bodies in olfactory learning[35]. Integration windows provide a flexible mechanism for maintaining sparseness; for example, the duration of the window can adjust to allow consistent responses to varying input conditions[36], increasing the dynamic range of sensory systems. Theoretical studies suggest coincidence detection with integration windows can enable more robust stimulus representations than coincidence detection with high spiking thresholds[37]. Other phase-specific effects of oscillations have been reported previously: work on the vertebrate brain performed *in vivo* and *in vitro*[12,13,38] as well as simulations[14,15] have shown that inputs arriving during the depolarizing phase of an oscillatory membrane potential can trigger action potentials with timing shaped by the oscillations. Discoveries of odour-evoked oscillations and phase-locked spiking in several insects, including bees[5], moths[6] and *Drosophila*[7], as well as in the mammalian olfactory cortex[39], suggest the principles revealed here are likely to apply throughout the animal kingdom.

## Methods

**Animals and preparation.** All experiments were performed on restrained, unanesthetized locusts, *Schistocerca americana*, raised in our crowded colony (hundreds of animals per cage), with 12-h dark, 12-h light cycle. Two-month-old animals ($n = 11$) of either sex were used in the experiments. Animals were immobilized and the brains were exposed, desheathed, and superfused with locust saline at room temperature as described previously[40].

**Odour delivery.** In some experiments odour pulses were used to elicit neural oscillations. Twenty millilitres of odorant solution (hexanol or cyclohexanone) was placed in 60 ml glass bottles at dilutions of 10% v/v in mineral oil. Odours, drawn from the headspace above these odorants, were puffed by a pneumatic picopump (WPI) into the continuously flowing air stream directed at the animal's antenna, and a large vacuum funnel behind the animal rapidly evacuated the odours, as described previously[41].

**Local field potential (LFP) recording.** To monitor odour-elicited oscillations, the LFP was recorded in the calyx of the mushroom body using gold-plated nickel-chrome wire (Kanthal). The signal was preamplified, amplified and filtered by a commercial amplifier (A-M Systems, Model 3600).

**Sharp intracellular recording.** Intracellular recordings were made from KC somata using sharp glass micropipettes (150–300 MΩ) filled with 0.2 M potassium acetate. The signals were amplified in bridge mode (Axoclamp-2B; Molecular Devices), further amplified with a DC amplifier (BrownLee Precision), and digitally sampled at 20 kHz (LabView software; USB-6353 DAQ card; National Instruments). KC somata could be targeted unambiguously above the mushroom body calyx[4].

**Intracellular stimulation.** Brief current pulses (5-ms duration) were injected into the cells through the intracellular electrode, using the 'step command' of the Axoclamp-2B amplifier. The timing of the pulses was controlled by a custom

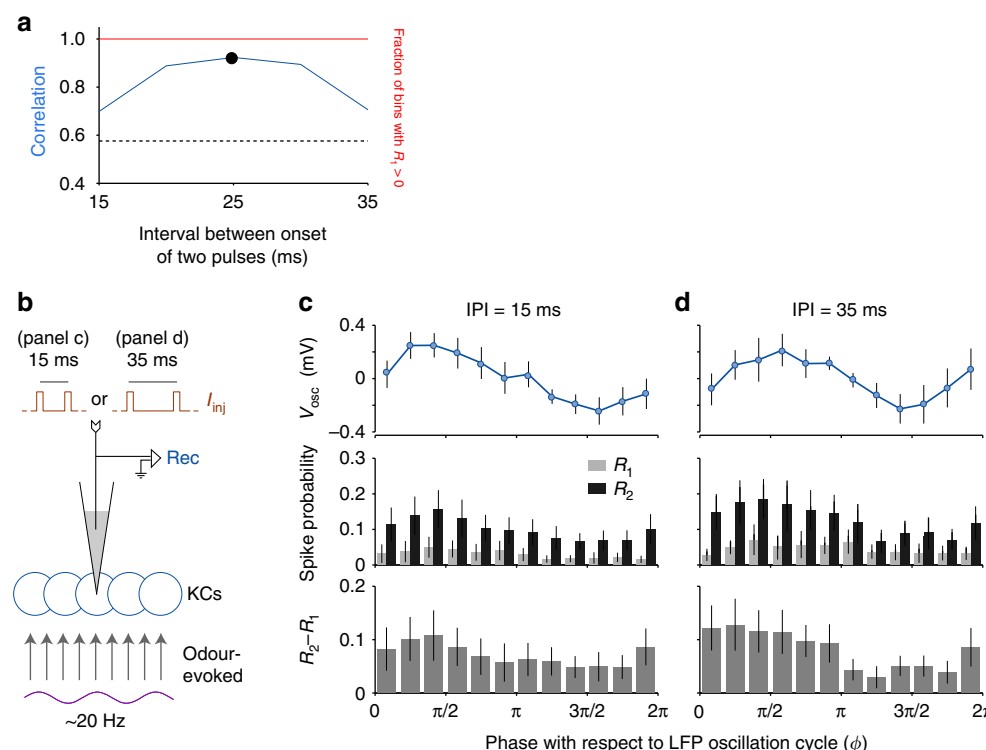

**Figure 6 | Integration windows are robust to variations in the inter-stimulus interval.** (**a**) The computational model predicted phase-dependent summation for a range of intervals between the two pulse inputs. (**b**) *In vivo* test: the experiment shown in Fig. 1, including odour-elicited oscillations, was repeated with two different inter-pulse intervals (IPIs); LFP electrode is not shown. (**c**) For 15-ms IPI, subthreshold oscillations and the effect of summation are phase-locked and correlated ($n = 18$ KCs). (**d**) The same result was obtained for 35-ms IPI ($n = 13$ KCs). Error bars represent s.e.m.

LabView program. The amplitude of the current pulses (typically between 0.1 and 0.2 nA) was set for each cell to elicit spikes on a fraction of the pulses.

**Oscillatory current injections.** To inject oscillatory current directly into the cell, we designed sinusoidal depolarizing waveforms in LabView and injected them using the analog output of the DAQ card. The output was amplified and low-pass filtered (1,000 Hz; BrownLee Precision) to remove high-frequency noise, de-amplified 1:1,000 using a custom voltage divider, and sent to the electrode using the DC-coupled 'external command' of the Axoclamp-2B amplifier. The peak-to-peak amplitude of the injected oscillatory current was 4 or 6 pA. The period of the injected waveform was either 50 ms (to mimic odour-evoked oscillations) or 84 ms (an arbitrarily chosen period that is harmonically unrelated to the 50-ms period, and is outside the range of frequencies normally evoked by odours[6]; a multiple of 12 was chosen to simplify binning into 12 bins; we did not test any other period). Pairs of current pulses were injected on top of the oscillatory input using the method described above. Pulses within a pair were separated by 25 ms. Pairs were separated by 122 and 133 ms for experiments with 50-ms and 84-ms waveforms, respectively; these times were long enough to allow the membrane potential to return to baseline before the next pair was delivered, and ensured that neighbouring pairs within a trial occurred at different phases of the oscillation cycle. Phases were binned relative to the oscillations in the membrane potential ($\pi/2$ denotes the peak of the membrane potential oscillation). Oscillatory current injections started 1-s before pulse injections (Fig. 4a); this allowed us to estimate the phase difference between injected current oscillations and resultant oscillations in the membrane potential, and use it for computing the phases of subsequent current pulses.

**Data analysis.** All analyses were performed using custom programs written in MATLAB (MathWorks). Spikes were detected by an algorithm and manually checked (blind to the LFP) to correct errors caused by superpositions with current-injection artefacts. Phases of spikes and current pulses (computed at pulse-offset) with respect to oscillations were estimated by measuring their positions relative to the previous (0 radian) and the next ($2\pi$ radians) peak in the simultaneously recorded LFP (noncausal band-pass filtered, 15–30 Hz). Current pulses occurring during asymmetrical LFP cycles were discarded if the peak-to-peak amplitude of the cycle was <25% of the maximum in that trial, or if the phase difference between two pulses in a pair deviated by more than $\pi/3$ from the expected phase difference (for example, $\pi$ for pulses separated by 25 ms). Spikes whose peaks occurred within a 6-ms window following the offset of a current pulse were considered to be triggered by the pulse. Spikes that occurred at least 20 ms after the end of a pair of pulses and before the beginning of the next pair were used for computing the phase-preference of odour-evoked spikes. Membrane potential oscillations were determined by estimating the average membrane potential, as a function of LFP phase (12 bins), in the 20-ms period (excluding periods that followed spikes) before the beginning of each pair of pulses. The peak-to-peak amplitude of oscillations was estimated for each cell from this binned waveform. The membrane properties of KCs were characterized by measuring their responses to current pulses and fitting them to exponential rise and decay curves. The amplitude of noise in vivo was estimated as the standard deviation in the membrane potential in the first 2-s of each trial, before any odour or current stimulus was provided (trials which had spikes during this period were excluded). The peak-to-peak noise amplitude was estimated as the difference between the maximum and the minimum value of the membrane potential in the 2-s period.

**Statistics.** We used two-tailed $t$ tests and Pearson correlations to make statistical comparisons. The sample sizes for these tests are specified by the number of bins in a cycle, which was predetermined to be 12. Thus, for correlation tests, $n = 12$; for two-sample $t$ tests, in which we compare the values in the first six bins and the last six bins, $n = 6$ per sample. To obtain reliable averages for each bin, we recorded from as many cells as we could (minimum of 10) from each animal in this study. Because the values in each phase bin are averages from multiple cells (between 10 and 18), their distribution is approximately normal (central limit theorem).

**Simulations.** A KC was modelled as a leaky integrate-and-fire neuron: $C\, dV/dt = -(V - E)/R + I$, where membrane capacitance $C = 10$ pF, and membrane resistance $R = 1,000$ MΩ (refs 4,33). The choice of the model was driven by its simplicity, small number of parameters and ease of comparison with experimental data. Simulations were performed in MATLAB, using 1/12 ms time-steps. If membrane potential $V$ exceeded the spiking threshold ($-41$ mV), a spike was generated and $V$ was reset to $E$ ($-65$ mV). A noise term was drawn every 1 ms from a uniform distribution between $-2$ and 2 mV, and added to $V$; the resulting cumulative noise distribution appears Gaussian. The LFP was represented by the sinusoidal wave $1 + \cos(2\pi t/T)$, with period $T = 50$ ms to mimic odour-evoked oscillations. The input to the cell, $I$, was modelled by the same excitatory sinusoidal wave (peak-to-peak amplitude = 5 pA), with a small time lag (6 ms) relative to LFP to simulate synaptic and conduction delays. Oscillations generated by this input in the membrane potential ($V_{osc}$) were further delayed because they reflect the integration of $I$. Two brief current pulses, $I_{inj}$ (amplitude = 0.05 nA, width = 5 ms,

inter-pulse interval = 25 ms) were added to $I$ at different phases of LFP in different simulation trials (96,000 trials in total). For this analysis, to exclude any role of after-hyperpolarization in generating phase-dependent summation, we included only those pulse pairs in which the first pulse did not elicit a spike.

**Simulations with uniform noise.** In biological neurons as well as in the model, accumulation of noise over time necessarily makes the distribution of noise non-uniform, regardless of the source of the noise. Therefore, to obtain uniform noise for one analysis, we used an unrealistic version of the model: the noise term was added to $V$ at every time-step during the pulses, for comparison with the spike threshold, but was removed from $V$ before going to the next time-step; this is equivalent to adding noise to the threshold and circumvents accumulation of noise. The noise term was drawn from a uniform distribution between $-5$ and 5 mV. Note that the duration of a pulse contains multiple simulation time steps; if different noise terms are drawn from a uniform distribution for different time-steps, the probability of spiking during a pulse will depend on the maximum of all the noise terms, which would deviate from the uniform distribution. To avoid this problem, we added the same noise term to all time-steps within a pulse.

**Data availability.** Data sets generated during this study are available from the corresponding authors on request.

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

## Acknowledgements

We thank Drs Maxim Bazhenov and Joby Joseph, and members of the Stopfer laboratory for helpful discussions, George Dold of the NIMH Instrumentation Core Facility for help with equipment, and Dr Kui Sun for her excellent animal care. This work was supported by an intramural grant from NIH-NICHD to M.S., and by the Department of Biotechnology in India (BT/08/IYBA/2014-8) and the Intermediate Fellowship of the Wellcome Trust—DBT India Alliance (IA/I/15/2/502091), to N.G.

## Author contributions

Conceptualization: N.G. and M.S.; Methodology: N.G. and M.S.; Investigation: N.G. and S.S.S.; Writing—Original Draft: N.G. and M.S.; Writing—Review and Editing: N.G. and M.S.; Funding Acquisition: N.G. and M.S.; Supervision: N.G. and M.S.

## Additional information

**Competing financial interests:** The authors declare no competing financial interests.

