## [Peer Review File · Nature Communications]

REVIEWERS' COMMENTS:

Reviewer #1 (Remarks to the Author):

The manuscript has markedly improved in the clarity of presentation both in text and figures. Also, the additional simulations are very, very helpful in appreciating the complexity of the phenomenon. I now find it a rewarding manuscript to read and the results compelling.

While there are extreme cases (as highlighted by the authors in the rebuttal letter and in the revised ms) where the result is "trivial" the authors clearly show that in general the effect is highly dependent on noise distributions and "non-trivial". To some extent this distinction is meaningless - I now understand the finding and the explanation so it might seem a "trivial" effect but that doesn't take away anything from the manuscript - in contrast.

Point-By-Point Response to Issues Raised by Referees

The final referee provided the following comments, reprinted here in full:

Reviewer #1 (Remarks to the Author):

The manuscript has markedly improved in the clarity of presentation both in text and figures. Also, the additional simulations are very, very helpful in appreciating the complexity of the phenomenon. I now find it a rewarding manuscript to read and the results compelling.

While there are extreme cases (as highlighted by the authors in the rebuttal letter and in the revised ms) where the result is "trivial" the authors clearly show that in general the effect is highly dependent on noise distributions and "non-trivial". To some extent this distinction is meaningless - I now understand the finding and the explanation so it might seem a "trivial" effect but that doesn't take away anything from the manuscript - in contrast.

We are pleased that the referee is now satisfied by our work.